# Association of Oxidized Low-Density Lipoprotein in Nonalcoholic Fatty Liver Disease with High-Risk Plaque on Coronary Computed Tomography Angiography: A Matched Case–Control Study

**DOI:** 10.3390/jcm11102838

**Published:** 2022-05-17

**Authors:** Takahiro Nishihara, Toru Miyoshi, Keishi Ichikawa, Kazuhiro Osawa, Mitsutaka Nakashima, Takashi Miki, Hiroshi Ito

**Affiliations:** 1Department of Cardiovascular Medicine, Faculty of Medicine, Dentistry and Pharmaceutical Sciences, Okayama University, Okayama 700-8558, Japan; taka.0204.hiro@gmail.com (T.N.); ichikawa1987@gmail.com (K.I.); mitsn1023@gmail.com (M.N.); tm.f20c.2000@gmail.com (T.M.); itomd@md.okayama-u.ac.jp (H.I.); 2Department of General Internal Medicine 3, Kawasaki Medical School General Medicine Centre, Okayama 700-0821, Japan; rohiwasa@yahoo.co.jp

**Keywords:** low-density lipoprotein cholesterol, nonalcoholic fatty liver disease, coronary computed tomography angiography, high-risk plaque, oxidized lipoprotein

## Abstract

Nonalcoholic fatty liver disease (NAFLD) is a risk factor for the development of atherosclerotic cardiovascular diseases (CVDs), and oxidative stress has been proposed as a shared pathophysiological condition. This study examined whether oxidized low-density lipoprotein (LDL) is involved in the underlying mechanism that links coronary atherosclerosis and NAFLD. This study included 631 patients who underwent coronary computed tomography angiography (CTA) for suspected coronary artery disease. NAFLD was defined on CT images as a liver-to-spleen attenuation ratio of <1.0. Serum-malondialdehyde-modified LDL (MDA-LDL) and coronary CTA findings were analyzed in a propensity-score-matched cohort of patients with NAFLD (*n* = 150) and those without NAFLD (*n* = 150). This study analyzed 300 patients (median age, 65 years; 64% men). Patients with NAFLD had higher MDA-LDL levels and a greater presence of CTA-verified high-risk plaques than those without NAFLD. In the multivariate linear regression analysis, MDA-LDL was independently associated with NAFLD (β = 11.337, *p* = 0.005) and high-risk plaques (β = 12.487, *p* = 0.007). Increased MDA-LDL may be a mediator between NAFLD and high-risk coronary plaque on coronary CTA. Increased oxidative stress in NAFLD, as assessed using MDA-LDL, may be involved in the development of CVDs.

## 1. Introduction

Nonalcoholic fatty liver disease (NAFLD) is a common condition with rising proportions in recent years, with a prevalence of 20–30% in the general population and 70–90% in obese or diabetic patients [1]. While NAFLD primarily affects liver structure and function, leading to morbidity and mortality from liver failure, cardiovascular diseases (CVDs) are the most common cause of death in patients with early NAFLD [2]. Moreover, emerging evidence has demonstrated that NAFLD is a risk factor for developing atherosclerotic cardiovascular complications, such as stroke and myocardial infarction [3,4,5]. We previously reported a significant association between NAFLD and the presence of high-risk plaques (HRP) on coronary computed tomography angiography (CCTA), which increased the likelihood of cardiovascular events [6]. However, the underlying mechanisms of increased cardiovascular diseases in patients with NAFLD have not been fully understood. Atherosclerosis and NAFLD co-occur in patients with metabolic syndrome, obesity, or type 2 diabetes mellitus. Therefore, it is difficult to determine the exact causal relationship that leads to an increased risk of CVD in patients with NAFLD.

Oxidative stress is one of the main factors associated with obesity and related disorders, such as cardiovascular diseases, metabolic syndrome, and type 2 diabetes mellitus, as well as cancer and neurodegenerative disorders. Oxidative stress is a phenomenon caused by an imbalance between production and accumulation of reactive oxygen species (ROS) in cells and tissues and the ability of a biological system to detoxify these reactive products. Among the mechanisms linking CVD risk with NAFLD, increased oxidative stress may represent a shared pathophysiological link between the two conditions. Oxidative stress plays a crucial role in the initiation and progression of atherosclerosis [7]. Excessive production of ROS is responsible for the oxidation of low-density lipoproteins (LDLs), which may promote the transformation of macrophages into foam cells, the first step in the formation of atherosclerotic lesions. In many clinical studies, elevated systemic markers of oxidative stress and lipid peroxidation have been observed in patients with NAFLD [8,9].

Herein, we hypothesized that increased oxidative LDL levels in patients with NAFLD are involved in the development of coronary atherosclerosis. In this study, we first investigated whether the serum levels of malondialdehyde-modified LDL (MDA-LDL), an oxidatively modified LDL [10], were increased in patients with NAFLD and then evaluated whether the increased MDA-LDL was associated with the presence of CCTA-verified HRP and NAFLD. 

## 2. Materials and Methods

### 2.1. Study Population

This retrospective, single-center, observational study was performed at Okayama University Hospital, Japan. Figure 1 shows a flow diagram of the study design. This study enrolled 3523 Japanese outpatients without a history of CAD who underwent CCTA for suspected CAD between August 2011 and December 2019. Patients were excluded if they had no data on MDA-LDL (*n* = 2842), consumed >20 g of alcohol per day (*n* = 19), had known liver diseases (carriers of hepatitis B or C virus, *n* = 17), or were using oral corticosteroids (*n* = 14). Finally, 631 patients were included in the study, of whom 165 were diagnosed with NAFLD by abdominal CT. Of the 165 patients with NAFLD, 150 age- and sex-matched patients were selected as the NAFLD group. Of the 466 non-NAFLD patients, 150 age- and sex-matched patients were included in the non-NAFLD group. The study protocol was approved by the institutional review board of Okayama University Hospital, and the study adhered to the Declaration of Helsinki. The requirement for informed consent was waived.

### 2.2. Anthropometric and Biochemical Measurements

The patients were checked for height, weight, drug and alcohol history, and other medical history through history taking and physical examinations. Complete blood counts, liver function tests, and biochemical analyses (total cholesterol, triglycerides, high-density lipoprotein (HDL) cholesterol, LDL cholesterol, MDA-LDL, etc.) were performed [11].

### 2.3. Acquisition of Coronary CTA

CT scans were performed using a 128-slice CT scanner (SOMATOM Definition Flash; Siemens Medical Solutions, Erlangen, Germany) as previously described [12]. All patients arrived at the hospital 1 h before the scheduled CT. If their heart rate was >60 beats/min, the patients received an oral beta-blocker. Patients mandatorily received an oral dose of short-acting nitroglycerin. The data were evaluated using a dedicated workstation (AZE Virtual Place; Canon Medical Systems Corporation, Otawara, Japan). The CCTA images were reconstructed with a slice thickness of 0.625 mm. With CCTA analysis, we evaluated coronary artery segments with a diameter >2 mm and defined plaque characteristics in accordance with the Society of Cardiovascular Computed Tomography [13].

### 2.4. Coronary Plaque Quantification

Each coronary segment was evaluated for the degree of stenosis (minimal, 1–29%; mild, 30–49%; moderate, 50–69%; severe, 70–99%; obstruction, 100% diameter stenosis). Stenosis was defined as significant if there was a luminal narrowing of ≥70% in any main coronary artery. We defined high-risk plaque (HRP) features (positive remodeling, spotty calcification, low-attenuation plaque) as previously described [14]. We defined positive remodeling as a remodeling index >1.1. Plaques with a CT attenuation number <30 HU were defined as low-attenuation. Spotty calcification was defined as a calcium burden length <1.5 times the vessel diameter and a width less than two-thirds of the vessel diameter. The presence of two or more of these features indicated an HRP.

### 2.5. Assessment of NAFLD

An abdominal non-contrast CT scan was performed immediately before the cardiac scan on the same day, as previously described [3]. Hepatic and splenic Hounsfield attenuations were measured using the maximum circular regions of interest in the liver and spleen (at least 1 cm^2^) [15]. The regions of interest included two areas that were aligned to the anteroposterior dimension of the right liver lobe and one that was aligned to the spleen. The hepatic-to-spleen attenuation ratio was calculated using mean Hounsfield unit (HU) measurements of the two right liver lobe regions of interest. A hepatic-to-spleen attenuation ratio of <1.0 was defined as the cutoff for a positive diagnosis of hepatic steatosis [16]. NAFLD was finally diagnosed after ruling out other causes of hepatic steatosis.

### 2.6. Definition of Risk Factors

The definition of risk factors has been previously described [17]. Diabetes mellitus was defined as a hemoglobin A1c ≥ 6.5% or the use of diabetic medications. Hypertension was defined as systolic blood pressure > 140 mmHg, diastolic blood pressure > 90 mmHg, or the use of antihypertensive drugs. Dyslipidemia was defined as one or more of the following: a fasting total cholesterol ≥ 240 mg/dL, LDL cholesterol ≥ 130 mg/dL, high-density lipoprotein cholesterol < 40 mg/dL, triglycerides ≥ 150 mg/dL, or current treatment with a lipid-lowering drug. Smoking status was defined as current smoking or non-smoking. Obesity was defined as a body mass index ≥ 25 kg/m^2^.

### 2.7. Statistical Analysis

Normality testing for continuous variables was performed using the Shapiro–Wilk test. Continuous variables are represented as mean ± standard deviation (SD) or median (interquartile range (IQR)) according to distribution. The model-estimated values are given as the means with 95% confidence intervals (CI). Categorical variables are presented as numbers (*n*) and percentages (%). We used a propensity-score-weighted analysis to adjust for potential confounding factors that may predispose patients to NAFLD and MDA-LDL. Propensity scores were calculated using a logistic regression model that included age and sex. Continuous variables were compared using the paired Student’s *t*-test or Mann–Whitney U-test, and categorical variables were compared using chi-squared analysis or Fisher’s exact test. Univariate and multivariate linear regression analyses were performed to determine the predictive factors for MDA-LDL. A *p*-value < 0.05 was considered statistically significant. All statistical analyses were performed using SPSS software (version 24; IBM Corp., Armonk, NY, USA) and R statistical package (version 3.5.2; R Foundation for Statistical Computing, Vienna, Austria).

## 3. Results

### 3.1. Patient Characteristics

After adjusting for age and sex, the median age of the study population was 65 years, and 193 (64%) patients were men. The prevalence rates of current smoking, hypertension, dyslipidemia, and diabetes mellitus among the participants were 20%, 64%, 59%, and 42%, respectively. The baseline characteristics of the patients with and without NAFLD are shown in Table 1. Patients with NAFLD had a higher prevalence of hypertension, dyslipidemia, diabetes mellitus, and obesity. The use of calcium channel blockers and oral antihyperglycemic drugs was more frequent among patients with NAFLD. Patients with NAFLD also had higher levels of glycated hemoglobin A1c (*p* < 0.001), C-reactive protein (*p* = 0.044), and brain natriuretic peptide (*p* = 0.001) than those without NAFLD. Of all, the median levels of LDL cholesterol and MDA-LDL were 110.0 mg/dL and 95.0 mg/dL, respectively. Patients with NAFLD had higher MDA-LDL levels than those without NAFLD (*p* = 0.001), whereas LDL cholesterol levels did not differ between the two groups. Patients with NAFLD also had higher triglyceride levels (*p* < 0.001) and lower HDL levels (*p* < 0.001) than those without NAFLD. The presence of significant stenosis in both groups was similar, whereas the presence of CTA-verified HRP in patients with NAFLD was higher than that in patients with NAFLD (*p* = 0.035).

### 3.2. Association between MDA-LDL and NAFLD

As shown in Table 2, the univariate logistic analysis demonstrated that MDAL-LDL was significantly associated with dyslipidemia (*p* = 0.003), CTA-verified HRP (*p* = 0.001), and NAFLD (*p* < 0.001). Factors and medications (obesity, hypertension, diabetes mellitus, calcium channel blockers, and oral antihyperglycemic drugs) were significantly associated with MDA-LDL. Multivariate linear regression analysis, including significant variable (*p*-value < 0.05) in the univariate analysis (dyslipidemia, high-risk plaque, and NAFLD), revealed that MDA-LDL was independently associated with dyslipidemia (*p* = 0.027), NAFLD (*p* = 0.005), and CTA-verified HRP (*p* = 0.007).

### 3.3. Association between Nonalcoholic Fatty Liver Disease and High-Risk Plaque

To evaluate the role of MDA-LDL in the presence of HRP in patients with NAFLD, logistic regression analysis was performed (Table 3). In univariate logistic regression analysis, the significant determinants of CTA-verified HRP were age, male sex, hypertension and dyslipidemia, use of ACE-I or ARB, use of oral antihyperglycemic drugs, NAFLD, and MDA-LDL. Furthermore, multivariate logistic regression analysis including significant variables (*p*-value < 0.05) in the univariate analysis (age, male sex, hypertension, dyslipidemia, use of ACE-I or ARB, use of oral antihyperglycemic drugs, NAFLD, and MDA-LDL) was performed. Multivariate logistic analysis revealed that age (odds ratio [OR], 1.033; 95% CI, 1.006–1.061) and MDA-LDL (OR, 1.011; 95% CI, 1.002–1.019) were independently associated with HRP, while the association between NAFLD and HRP was not significant.

## 4. Discussion

In this study, we hypothesized that oxidized LDL is involved in the development of CVD in patients with NAFLD. We demonstrated that MDA-LDL, a marker of oxidative stress, is independently associated with the presence of NAFLD and CTA-verified HRP. Additionally, the association between NAFLD and CTA-verified HRP was not significant when MDA-LDL was included as a confounding factor. These findings suggest an interplay between increased MDA-LDL, NAFLD, and CTA-verified HRP (Figure 2).

Among the mechanisms linking CVD risk with NAFLD, the most prominent are insulin resistance, low-grade chronic inflammation, and atherogenic dyslipidemia [18]. In addition, increased oxidative stress may represent a shared pathophysiological condition between the two conditions [7]. This study showed that serum levels of MDA-LDL, which is produced by LDL oxidation, were significantly higher in patients with NAFLD than in those without NAFLD. When fat accumulates in hepatocytes and the cholesterol content in the mitochondrial membrane increases, ROS production from mitochondria is enhanced. Lipid peroxidation in the liver can induce the progression of NAFLD to nonalcoholic steatohepatitis by stimulating Kupffer cells, hepatic stellate cells, and hepatocytes [8]. Thus, oxidative stress may be considered one of the main contributors involved in the development and risk of NAFLD progression to nonalcoholic steatohepatitis characterized by inflammation and fibrosis. Meanwhile, several studies showed that systemic oxidative stress parameters, including oxidative LDL in patients with NAFLD, were increased compared with patients without NAFLD [19,20]. This is an important observation as oxidative LDL is potently atherogenic and is very important in the development of coronary artery disease. Oxidative LDL is directly involved in the initiation and progression of the atherosclerotic disease process, from the early-stage conversion of macrophages into foam cells to the late-stage development of coronary artery stenosis, plaque rupture, and coronary thrombosis [21]. Based on the finding that MDA-LDL is independently associated with the presence of NAFLD and CTA-verified HRP, increased MDA-LDL may be a mediator between NAFLD and CTA-verified HRP.

MDA-LDL is taken up by macrophages under the blood vessels to form foam cells that accumulate a large amount of cholesterol and impair the function of endothelial and smooth muscle cells [22]. Macrophages incorporating MDA-LDL produce and release ROS and inflammatory mediators such as tumor necrosis factor-α, interleukin-6, and interleukin-1. The presence of systemic inflammation promoted by cytokines leads to endothelial dysfunction, altered vascular tone, and enhanced vascular plaque formation. Inflammation due to oxidative stress plays an important role in the onset and progression of arteriosclerosis and vulnerable plaque formation [21].

We have reported that NAFLD is associated with an increased risk of cardiovascular events compared with the population without NAFLD [3,5]. In these studies, the CTA-verified HRP showed a substantial impact on the incidence of CVD in patients with NAFLD. Recent advances in CCTA can visualize coronary plaques with positive remodeling, spotty calcification, and low-attenuation plaques; these characteristics are associated with a high risk of acute coronary syndrome [23]. In line with our previous data [6], this study showed that the presence of CTA-verified HRP in patients with NAFLD was significantly greater than that in those without NAFLD. Considering the significant association between MDA-LDL levels and CTA-verified HRP, serum MDA-LDL is a potent risk factor for CVD in patients with NAFLD. However, clinical outcome data have not yet been obtained. Therefore, the clinical impact of MDA-LDL in patients with NAFLD should be evaluated in future studies.

This study has some limitations. Firstly, our study was cross-sectional; thus, we could not establish a causal relationship between NAFLD, MDA-LDL, and CTA-verified HRP. Secondly, this study was conducted retrospectively at a single center with a limited number of patients. In addition, our study population only consisted of Japanese patients with suspected CAD, and the results therefore cannot be applied directly to other ethnic groups. Thirdly, CT results may not be sufficient for the diagnosis of NAFLD. The histological data supporting the diagnosis of NAFLD were not documented in our study. Fourthly, we did not exclude patients who underwent statin therapy. The inclusion of these patients may have affected our results.

## 5. Conclusions

We demonstrated that MDA-LDL, a marker of oxidative stress, is independently associated with the presence of NAFLD and CTA-verified HRP. Increased oxidative stress in NAFLD, as assessed using MDA-LDL, may be involved in the development of CVD. Further studies are needed to evaluate the association between MDA-LDL levels and the incidence of CVD events.

## Figures and Tables

**Figure 1 jcm-11-02838-f001:**
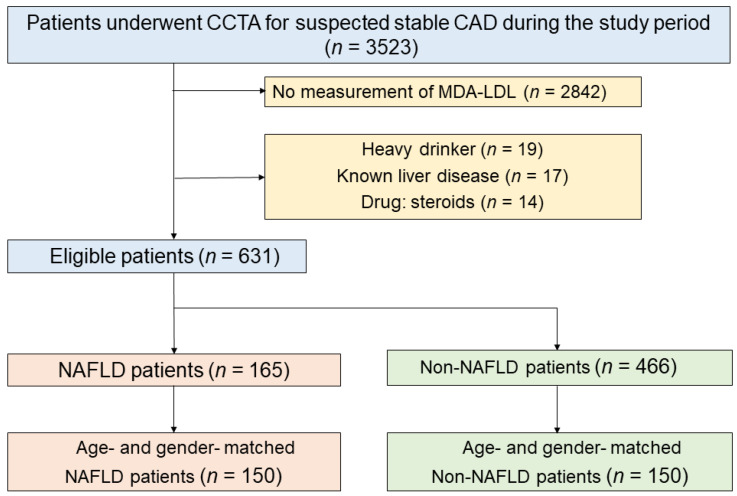
Flow diagram of the study.

**Figure 2 jcm-11-02838-f002:**
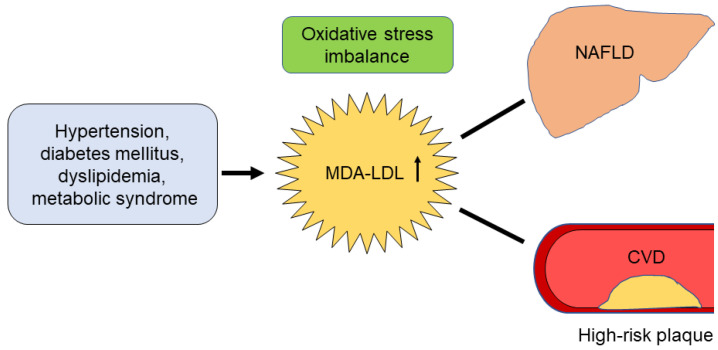
Possible pathological mechanisms linking nonalcoholic fatty liver disease and cardiovascular disease. MDA-LDL, malondialdehyde-modified low-density lipoprotein; NAFLD, nonalcoholic fatty liver disease; CVD, cardiovascular disease.

**Table 1 jcm-11-02838-t001:** Patients’ characteristics.

	All (*n* = 300)	NAFLD		*p*-Value
		Present (*n* = 150)	Absent (*n* = 150)	
Age, years	65 (55, 71)	65 (55, 70)	65 (55, 71)	0.734
Male gender	193 (64)	96 (64)	97 (65)	0.904
Hypertension	192 (64)	105 (70)	87 (58)	0.030
Dyslipidemia	177(59)	97 (65)	80 (53)	0.046
Diabetes mellitus	125 (42)	73 (49)	52 (35)	0.014
Current Smoker	61 (20)	27 (18)	34 (23)	0.315
Obesity *	141 (48)	100 (68)	41 (28)	<0.001
Beta blocker	72 (24)	40 (27)	32 (21)	0.279
Calcium channel blocker	98 (33)	60 (40)	38 (25)	0.007
ACE-I or ARB	98 (33)	52 (35)	46 (31)	0.460
Statin	124 (41)	64 (43)	60 (40)	0.639
Oral antihyperglycemic drugs	80 (27)	50 (33)	30 (20)	0.009
eGFR, mL/min/1.73 m^2^	73.2 (63.3, 83.7)	72.0 (62.1, 83.1)	73.7(64.4, 84.3)	0.445
AST, IU/L	22 (17, 28)	24 (19, 31)	20 (17, 25)	<0.001
ALT, IU/L	21 (15, 31)	26 (19, 39)	17 (12, 23)	<0.001
HbA1c, %	6.1 (5.7, 6.8)	6.4 (5.8, 7.2)	5.9 (5.6, 6.5)	<0.001
CRP	0.09 (0.05, 0.17)	0.11 (0.06, 0.20)	0.08 (0.04, 0.17)	0.044
BNP	25.5 (13.0, 53.4)	19.8 (10.5, 44.4)	28.9 (15.3, 81.9)	0.001
Total cholesterol, mg/dL	186.8 ± 39.7	186.4 ± 37.1	187.2 ± 42.1	0.860
HDL cholesterol, mg/dL	53.0 (44.0, 66.3)	50.5 (43.0, 61.0)	58.0 (46.8, 72.0)	<0.001
Triglyceride, mg/dL	116.0 (86.0, 178.3)	133.0 (100.3, 202.3)	101.0 (77.8, 149.0)	<0.001
LDL cholesterol, mg/dL	110.0 (88.0, 132.0)	111.5 (92.0, 132.8)	106.5 (86.8, 129.3)	0.258
MDA-LDL, U/L	95.0 (74.3, 119.0)	100.5 (78.8, 127.3)	87.5 (71.0, 111.3)	0.001
High-risk plaque **	78 (26)	47 (31)	31 (21)	0.035
Significant stenosis ***	112 (37)	56 (37)	56 (37)	1.000

Values are expressed as the median (interquartile range), mean ± standard deviation, or number (%). * Obesity was defined as a body mass index ≥ 25 kg/m^2^. ** A high-risk plaque is defined by the presence of two or more features (positive remodeling, spotty calcification, and low-attenuation plaque). *** Stenosis is defined as a luminal narrowing of ≥70% in any coronary artery. ACE-I—angiotensin-converting enzyme inhibitor; ARB, angiotensin-receptor blocker; eGFR, estimated glomerular filtration rate; LDL, low-density lipoprotein; HDL, high-density lipoprotein; HbA1c, glycated hemoglobin A1c; CRP, C-reactive protein; BNP, brain natriuretic peptide; MDA-LDL, malondialdehyde-modified low-density lipoprotein.

**Table 2 jcm-11-02838-t002:** Factors associated with MDA-LDL.

	Univariate		Multivariate ***	
	β	*p*-Value	β	*p*-Value
Hypertension (yes, no)	4.596	0.286		
Dyslipidemia (yes, no)	12.249	0.003	9.111	0.027
Diabetes mellitus (yes, no)	−1.485	0.722		
Current Smoker (yes, no)	2.948	0.551		
Obesity * (yes, no)	7.726	0.060		
Beta-blocker (yes, no)	−4.353	0.375		
Calcium channel blocker (yes, no)	3.132	0.485		
ACE-I or ARB (yes, no)	−1.939	0.666		
Statin (yes, no)	−4.386	0.287		
Oral antihyperglycemic drugs (yes, no)	−1.795	0.705		
High-risk plaque ** (yes, no)	15.389	0.001	12.487	0.007
NAFLD (yes, no)	13.653	<0.001	11.337	0.005

* Obesity was defined as a body mass index ≥ 25 kg/m^2^. ** A high-risk plaque is defined by the presence of two or more features (positive remodeling, spotty calcification, and low-attenuation plaque). *** Multivariate linear regression analysis was performed using significant factors in the univariate analysis (*p*-value < 0.05). MDA-LDL, malondialdehyde-modified low-density lipoprotein; NAFLD, Nonalcoholic fatty liver disease.

**Table 3 jcm-11-02838-t003:** Factors associated with high-risk plaque.

	Univariate		Multivariate **	
	Hazard Ratio (95%CI)	*p*-Value	Hazard Ratio (95%CI)	*p*-Value
Age, years	1.037 (1.014–1.061)	0.002	1.033 (1.006–1.061)	0.016
Male gender (yes, no)	1.869 (1.051–3.322)	0.033	1.565 (0.826–2.965)	0.170
Hypertension (yes, no)	3.500 (1.820–6.731)	<0.001	1.830 (0.825–4.063)	0.137
Dyslipidemia (yes, no)	2.138 (1.211–3.773)	0.009	1.318 (0.691–2.515)	0.402
Diabetes mellitus (yes, no)	1.575 (0.934–2.657)	0.088		
Current Smoker (yes, no)	1.561 (0.844–2.885)	0.155		
Obesity * (yes, no)	1.252 (0.744–2.107)	0.397		
Statin (yes, no)	1.612 (0.958–2.711)	0.072		
Beta blocker (yes, no)	1.048 (0.570–1.929)	0.879		
Calcium channel blocker (yes, no)	1.591 (0.918–2.757)	0.098		
ACE-I or ARB (yes, no)	2.177 (1.258–3.769)	0.005	1.427 (0.737–2.763)	0.291
Oral antihyperglycemic drugs (yes, no)	1.909 (1.081–3.370)	0.026	1.503 (0.801–2.819)	0.204
NAFLD (yes, no)	1.752 (1.037–2.960)	0.036	1.523 (0.834–2.781)	0.171
MDA-LDL, per U/L	1.012 (1.005–1.019)	0.001	1.011 (1.002–1.019)	0.012

* Obesity was defined as a body mass index ≥ 25 kg/m^2^. ** Multivariate linear regression analysis was performed using significant factors in the univariate analysis (*p*-value < 0.05). ACE-I, angiotensin-converting enzyme inhibitor; ARB, angiotensin-receptor blocker; NAFLD, Nonalcoholic fatty liver disease; MDA-LDL, malondialdehyde-modified low-density lipoprotein.

## Data Availability

The data presented in this study are available upon request from the corresponding author. The data is not publicly available because of privacy concerns.

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
