# Peer review of "Association of Oxidized Low-Density Lipoprotein in Nonalcoholic Fatty Liver Disease with High-Risk Plaque on Coronary Computed Tomography Angiography: A Matched Case–Control Study"

_jcm, 2022, doi:10.3390/jcm11102838_

Round 1

Reviewer 1 Report

In general, it is a well written manuscript about the association of oxidative stress, NAFLD and high-risk coronary artery disease based on imaging with coronary CT. The authors have demonstrated in a precise way that oxidative stress might be a mediator for the development of high-risk CAD in patients with NAFLD. Nevertheless, I have the following minor comments to suggest:

  1. In the segment of methods, in the definition of stable angina, you should place the appropriate reference. 
  2. In the multivariate analysis, please specify which factors you used, which is the reference category if each variable and the p-values of the comparisons (provide a better presentation of the data in the manuscript in paragraph 3.2, 3.3 and in tables 2 and 3).
  3. In the segment of discussion, you could provide in more details the pathophysiologic associations between oxidative LDL and NAFLD as well as oxidative LDL and coronary atherosclerosis.

Author Response

Response to the reviewer #1:

We thank Reviewer #1 for giving us an opportunity to revise our manuscript. We revised our manuscript to address the recommendation.

Our point-by-point responses to each of the reviewer’s comments and suggestions are listed below. The reviewer’s unedited comments are in bold text, followed by our response to each comment. New text in the revised manuscript appears as red text.

In the segment of methods, in the definition of stable angina, you should place the appropriate reference. 

Response: We have double checked participants in this study. This study included both symptomatic and asymptomatic patients with suspected CAD. We apologize for the inappropriate description. We have deleted the following sentence.

Page 2 line 66-68 (former manuscript)

Stable CAD was defined as angina with no changes in the frequency, duration, or in-tensity of anginal symptoms within 4 weeks before CCTA.

In the multivariate analysis, please specify which factors you used, which is the reference category if each variable and the p-values of the comparisons (provide a better presentation of the data in the manuscript in paragraph 3.2, 3.3 and in tables 2 and 3).

Response: Multivariate linear regression analysis was performed using significant factors in the univariate analysis (p-value <0.05). We added this explanation in the manuscript in paragraph 3.2, 3.3 and in tables 2 and 3.

In the segment of discussion, you could provide in more details the pathophysiologic associations between oxidative LDL and NAFLD as well as oxidative LDL and coronary atherosclerosis.

Response: We added explained the pathophysiologic associations between oxidative LDL and NAFLD as well as oxidative LDL and coronary atherosclerosis as follows.

Page 7-line 254Page 8, line 267

When fat accumulates in hepatocytes and the cholesterol content in the mitochondrial membrane increases, oxygen reactive species production from mitochondria is enhanced. Lipid peroxidation in the liver can induce the progression of NAFLD to nonalcoholic steatohepatitis by stimulating Kupffer cells, hepatic stellate cells, and hepatocytes [8]. Thus, oxidative stress may be considered one of the main contributors involved in the development and risk of NAFLD progression to nonalcoholic steatohepatitis characterized by inflammation and fibrosis. Meanwhile, several studies showed that systemic oxidative stress parameters including oxidative LDL in patients with NAFLD were increased compared to patients without NAFLD [19,20]. This is an important observation as oxidative LDL is potently atherogenic and very important in the development of coronary artery disease. Oxidative LDL is directly involved in the initiation and progression of the atherosclerotic disease process, from the early-stage conversion of macrophages into foam cells to the late-stage development of coronary artery stenosis, plaque rupture, and coronary thrombosis [21].

Reviewer 2 Report

Nishihara et al observed that oxidative stress marker malondialdehyde-modified LDL is independently associated with the presence of NAFLD and CTA-verified HRP. The current study is interesting and relevant to the field of cardiovascular biology.

However, this study has a lot of limitations, which were mentioned by the author at the end of the discussion section. This reviewer has some minor concerns that could be addressed in the revised manuscript.

  1. Did the author check other oxidative stress markers apart from MDA-LDL?
  2. Since this study is based on the oxidized LDL (MDA-LDL) as an oxidative stress marker that increased in NAFLD, the author may need to reframe the title of the manuscript accordingly.
  3. The introduction section can be improved.

Author Response

Response to the reviewer #2:

We thank Reviewer #2 for giving us an opportunity to revise our manuscript. We revised our manuscript to address the recommendation.

Our point-by-point responses to each of the reviewer’s comments and suggestions are listed below. The reviewer’s unedited comments are in bold text, followed by our response to each comment. New text in the revised manuscript appears as red text.

Did the author check other oxidative stress markers apart from MDA-LDL?

Response: Thank you for your comments. We didn’t check other oxidative stress markers.

Since this study is based on the oxidized LDL (MDA-LDL) as an oxidative stress marker that increased in NAFLD, the author may need to reframe the title of the manuscript accordingly.

Response: We changed the title as follows

Title

 “Association of high oxidized low-density lipoprotein in non-alcoholic fatty liver disease with high-risk plaque on coronary computed tomography angiography: A matched case-control study”.

The introduction section can be improved.

Response: We have revised the introduction section as follows.

Introduction

   Nonalcoholic fatty liver disease (NAFLD) is a frequent condition with rising pro-portions in recent years, with a prevalence of 20%–30% in the general population and 70%–90% in obese or diabetic patients [1]. While NAFLD primarily affects liver structure and function, leading to morbidity and mortality from liver failure, cardiovascular diseases (CVDs) are the most common cause of death in patients with early NAFLD [2]. Moreover, emerging evidence has demonstrated that NAFLD is a risk factor for developing atherosclerotic cardiovascular complications, such as stroke and myocardial infarction [3–5]. We previously reported a significant association between NAFLD and the presence of high-risk plaques (HRP) on coronary computed tomography angiography (CCTA), which increased the likelihood of cardiovascular events [6]. However, the underlying mechanisms of the increased cardiovascular diseases in patients with NAFLD have not been fully understood. Atherosclerosis and NAFLD co-occur in patients with metabolic syndrome, obesity, or type 2 diabetes mellitus. Therefore, it is difficult to decipher the exact causal relationship that leads to an increased risk of CVD in patients with NAFLD.

   Oxidative stress is one of the main factors associated with obesity and related dis-orders, such as cardiovascular diseases, metabolic syndrome, type 2 diabetes mellitus, as well as cancer, and neurodegenerative disorders. Oxidative stress is a phenomenon caused by an imbalance between production and accumulation of oxygen reactive species (ROS) in cells and tissues and the ability of a biological system to detoxify these reactive products. Among the mechanisms linking CVD risk with NAFLD, increased oxidative stress may represent a shared pathophysiological condition between the two conditions. Oxidative stress plays a crucial role in the initiation and progression of atherosclerosis [9]. Excessive production of ROS is responsible for the oxidation of low-density lipoproteins (LDLs), which may promote the transformation of macro-phages into foam cells, the first step in the formation of atherosclerotic lesions. In many clinical studies, elevated systemic markers of oxidative stress and lipid peroxidation have been observed in patients with NAFLD [7,8].

   Herein, we hypothesized that increased oxidative LDL levels in patients with NAFLD are involved in the development of coronary atherosclerosis. In this study, we first investigated whether serum levels of malondialdehyde-modified LDL (MDA-LDL), an oxidatively modified LDL [10], were increased in patients with NAFLD and then evaluated whether the increased MDA-LDL was associated with the presence of CCTA-verified HRP and NAFLD.

This manuscript is a resubmission of an earlier submission. The following is a list of the peer review reports and author responses from that submission.